# Probiotic Characteristics and Antifungal Activity of *Lactobacillus plantarum* and Its Impact on Fermentation of Italian Ryegrass at Low Moisture

**Karnan Muthusamy [1,†], Ilavenil Soundharrajan [1,†], Srigopalram Srisesharam [1,†], Dahye Kim [2], Palaniselvam Kuppusamy [1], Kyung Dong Lee [3] and Ki Choon Choi [1,*]**

[1] Grassland and Forage Division, National Institute of Animal Science, Rural Development Administration, Chungcheongnam-do, Cheonan-si 31000, Korea; karnantm111@gmail.com (K.M.); arulvenil@rediffmail.com (I.S.); srigopal.ram@gmail.com (S.S.); kpalaselvamsmailbox@rediffmail.com (P.K.)

[2] Center for Research on Environmental Disease, College of Medicine, University of Kentucky, 1095 VA Drive, Lexington, KY 40536, USA; pioioiq10@gmail.com

[3] Department of Oriental Medicine Materials, Dongsin University, Naju 58245, Korea; leehj5328@korea.kr

\* Correspondence: choiwh@korea.kr; Tel.: +82-41-580-6752; Fax: +82-41-580-6779

† These authors contributed equally to this work.

**Abstract:** The study aimed to investigate probiotic characteristics, and low moisture silage fermentation capability of selected lactic acid bacteria (LAB) isolated from Alfalfa (*Medicago sativa* L). Morphological and physiological properties, carbohydrates fermentation, enzymes, and organic acids production, anti-fungal activity, antibiotic sensitivity patterns, and probiotic characteristics (acidic and bile salt tolerances, hydrophobicity and aggregations natures) of LAB were examined. 16SrRNA sequencing was carried out to identify isolated strains. The identified strains *Lactobacillus plantarum* (KCC-37) and *Lactobacillus plantarum* (KCC-38) showed intense antifungal activity, survival tolerant in acidic and bile salt environments, cell surface and auto aggregations ability, enzymes and organic acids productions. At ensiled condition, KCC-37 and KCC-38 enhanced acidification of Italian ryegrass silages by producing a higher amount of lactic acid, a key acid for indicating silage quality with less extent to acetic acid and succinic acid at low moisture level than non-inoculated silages. Notably, the addition of mixed strains of KCC-37 and KCC-38 more potentially enhanced acidification of silage and organic acid productions than the single-culture inoculation. The overall data suggested that these strains could be used as an additive for improving the quality of the fermentation process in low moisture silage with significant probiotic characteristics.

**Keywords:** *Lactobacillus plantarum*; antifungal; probiotic; Italian ryegrass; low moisture silage

## 1. Introduction

The livestock sector is a promising and rapid growing segment in the agriculture economy of the developing countries. For ruminants, roughage is the primary source of the diet; the quality of silage determines their growth, which brings economic benefits to farmers [1]. For that reason, more concentration has been given mainly to high-quality silage/haylage production [2]. Lactic acid bacteria (LAB) is considered as a major group of starter cultures with better performances and high competitiveness probiotics used in dairy products, meat, fish, fruit, vegetables and cereals products. Also, LAB contributes to the enhancement of the nutritional value of fermented foods, cheese maturation, and yoghurt texture with exopolysaccharide production and prevents the secondary fermentation process. LAB are proficient in preventing pathogenic bacterial growth through different mechanisms including adherence to cells wall, modulation of immune system and secretion of antimicrobial

agents [3,4]. Therefore, isolation and identification of effective probiotic LAB with survival and colonizing ability under gastrointestinal tract conditions [5] from fermented foods and other products have gained great attention in recent days.

Fungal spoilage and mycotoxin contaminations are one of the highest risks of stored feed such as silage. It affects nutrients composition, dry matter, palatability reduction and reduction in consumption of silage. The main fungi isolated from contaminated silages were *Aspergillus*, *Perncillum* and *Fusarium* genera followed by mucor, and others [6–8]. Homofermentative *Lactobacilli* includes *Lactobacilli casei* and *L. plantarum* that directly enhance the silage/haylage quality by preventing undesirable microbial growth via lowering surrounding pH. On the other hand, haylage quality was affected by *Enterococcus* strains like *Enterococcus hirae*, *E. faecalis*, *E. casseliflavus* and *E. mundtii*, which are broadly dispersed on forage crops [9]. The fermented forage pH and its butyric acid level were decreased rapidly by all the tested strains [10]. At 25 °C, the perennial rye-grass haylage showed good results by the *P. pentosaceus*, *P. acidilactici* and *L. plantarum* than *E. faecalis*, *E. faecium*, *E. casseliflavus*, *W. paramesenteroides* and *L. pseudomesenteroides*. But at 45 °C, only *P. acidilactici* showed a better effect on haylage fermentation. Those previous study results demonstrated that haylage fermentation was significantly improved by LAB species [11]. LAB and temperature play a major role in the fermentation process. The growth temperature of these bacteria broadly ranged from 5 °C to 50 °C, which shows that each LAB has its unique optimal growth temperature [12]. Differently, cold-tolerant LAB also produces lactic acid at low temperature that overcomes the trouble of deficient lactic fermentation.

In an anaerobic condition, lactic acid bacteria (LAB) utilize water-soluble carbohydrates (WSC) and converted into organic acids particularly lactic acid that gradually decrease pH the surround environment during the fermentation process, are known as ensiling [13]. Enhancement of lactic acid production is an essential factor for developing high quality silage and used to preserve the quality of silage for a long time. Several bacterial inoculants are commercially available and used to enhance the forages fermentation in the silo, showing that good fermentation that directly proportional to more numbers of lactic acid bacteria with the high level of lactic acid production. Among these, homolytic bacteria *lactobacillus* species have been considered as the most suitable strains in the enhancement of high-quality silage/haylage production [14]. This process of fermentation by adding *Lactobacillus* to haylage has plenty of positive effects like non-corrosive to farm machinery, less cost than enzyme formulation and no harm to the environment [15]. In general, different types of LAB species have been found in the plants and grass that naturally involved in the fermentation process, but it is not sufficient to rapid fermentation of silage/haylage and their nutrient contents. Therefore, adding efficient *lactobacillus* strains during ensiling is considered the most valuable aspects to develop good quality silage/haylage as feed ruminant animals. In this aspect, we isolated *Lactobacillus plantarum* strains from fermented alfalfa and investigated their probiotics features and antifungal efficiency. Finally, the impact of isolated strains on the fermentation of low moisture Italian ryegrass silages was studied.

## 2. Materials and Methods

### 2.1. Isolation and Preliminary Screening of Lactic acid Bacteria

The alfalfa (*Medicago sativa*) samples were collected from Agricultural farm, Rural Development Administration, Cheonan, Republic Korea. A one gram sample was diluted with 10 mL sterile distilled water, and then made serial dilution. Diluted samples were spread on De Man Rogosa and Sharpe Agar (MRS agar, CONDA, Madrid, Spain). Plates were incubated at 30 °C for 72 h. LAB colonies were picked up and streaked on bromo cresol purple agar medium (BCP, CONDA, Madrid, Spain) to confirm the LAB identity [2]. Different bacterial isolates were isolated and stored in 4 °C for short time storage. Based on growth profile analysis, antimicrobial results, biochemical and physiological analysis, potential strains were selected for further studies. An overnight culture of the bacterial strains used and performed with Quantification of fermentative acids (Megazyme assay kit—Bray, Co. Leinster, Wicklow, Ireland), enzyme production and carbohydrate fermentation (API-CH50 and

API-ZYM kits, Marcy-I' Etoile, France). The antibiotic sensitivity was screened by disc method [13]. The morphological characteristics of isolates were identified using Scanning Electron Microscopy [16] (SEM, JSM6460 model, JEOL Ltd., Peabody, MA, USA).

*2.2. Molecular Characterization*

The DNA of the KCC-37 and KCC-38 was extracted using the commercial kit method (QlAamp DNA mini kit-Qiagen, MD, USA). The 16S rRNA gene was amplified by PCR (Applied Bi system-9700, Thermofisher, Scientific, Waltham, MA, USA) using universal primers (27F AGA GTT TGA TCM TGG CTC AG and 1492R GTA TTA CCG CGG CTG CTG G) with genomic DNA as a template and its sequencing by Genetic Analyzer 3130 [17] (Applied Bio-system, Foster, CA, USA) in Solgent Co. Ltd. (Seoul, Korea). The aligned 16S rRNA of the isolate was subjected to BLAST with the non-redundant Gene Bank database [18]. Based on the maximum identity score was selected and aligned using ClustalW multiple alignment software.

*2.3. Antifungal Activity of KCC-37 and KCC-38*

2.3.1. Fungal and Cultures and Preparations

Fungal strains used in the study such as *Scopulariopsis brevicaulis* (KACC 40273), *Penicillium chrysogenum* (KACC 40399), *Aspergillus clavatus* (KACC 40071), *Aspergillus flavus* (KACC-40232) and *Fusarium oxysporum* (KACC 40051) were obtained from the Korean Agriculture Culture Collection (KACC), South Korea. All fungi were plated in PDA medium and harvested conidial spore using 0.1% Tween 20 in distilled water for further use [16].

2.3.2. Production of Extracellular Metabolites at Different Temperatures and pH Conditions

Fresh KCC-37 and KCC-38 were inoculated in MRS broth and incubated at different temperatures (25, 35, and 45 °C) for 48hrs. Second methods, KCC-37 and KCC-38 were inoculated in pH adjusted MRS broth (pH-5.5, 6.5 and pH-7.5) for 48 h. Cell-free metabolites (CFM) were collected from both experiments by centrifugation at 4000× $g$. The collected CFM was filtered and used for anti-fungal activity.

2.3.3. Antifungal Activity of KCC-37 and KCC-38

Pour Plate Method

Fresh KCC-37 and KCC-38 colonies were spotted on the MRS agar medium and incubated at 30 ± 2 °C for 24 h. Conidial fungi (*S. brevicaulis*, *P. chrysogenum*, *A. clavatus*, *A. flavus* and *F. oxysporum*) were suspended in 0.8% PD agar medium and laid over the MRS medium and incubated at 35 ± 2 °C for 72 h. The conidial growth was monitored every day with and without LAB inoculated plates. The zone inhibition around KCC-37 and KCC-38 colony was measured [16].

Fungal Biomass Inhibition

Cell-free metabolites (CFM) produced by KCC-37 and KCC-38 were diluted in PD broth (Conditional medium) at the concentration of 10%. Conidial spore of each fungus (*S. brevicaulis*, *P. chrysogenum*, *A. clavatus*, *A. flavus* and *F. oxysporum*) was inoculated in conditional medium and incubated at 35 ± 2 °C for 72 h. The fungal mate was collected and filtered using Whatman no 1 filter paper and dried in an oven for 48 h at 45 °C. The average fungal biomass was calculated for control and CFM treated samples. Fungi grown in 10% MRS broth in PD broth was considered as control [19].

*2.4. Survival in Low pH and Bile Salts Environment*

MRS broth at pH range of 2.5 was prepared with 1 M HCl. Fresh culture was inoculated into this media and incubated at 37 °C for 3 h and then LAB was enumerated. For bile salts resistant study;

fresh cultures were inoculated in MRS broth containing 0.3% oxgall and 0.5% sodium deoxycholate (SDC) bile salts (Sigma, St Louis, MO, USA) and incubated at 37 °C for 24 h. Bacterial survival rate was determined after 10 fold serial dilution on the BCP and MRS agar plates [20]. The results were expressed as the growth of the bacteria, as enumerated by agar plates.

### 2.5. Autoaggregation

Freshly grown KCC-37 and KCC-38 in MRS was centrifuged at 4000× *g* and 4 °C for 10 min and the collected pellet was washed twice with PBS and re-suspended in 4 mL of PBS and taken initial optical density (OD) at 600 nm and then incubated for 3 h at 37 °C. Aliquots (100 µL) withdrawn at a regular interval of one hour from the top of the upper part and mixed with 3.9 mL of PBS and read at 600 nm. The percentage of autoaggregation (AA) was calculated by the following equation. %AA $= 1 - (At/A0) * 100$ $A_t$ = absorbance at different time intervals (1, 2 and 3 h), $A_0$ = initial absorbance at 0 h [20].

### 2.6. Hydrophobicity of KCC-37 and KCC-38

The cell surface hydrophobicity property of KCC-37 and KCC-38 were analyzed by Rosenberg et al. with slight modifications [21]. Fresh KCC-37 and KCC-38 grown in MRS broth was centrifuged at 4000× *g* for 10 min and collected the pellet, and then washed with PBS twice. After washed, the pellets were suspended in PBS. Three ml of KCC-37 and KCC-38 suspension and 1 mL of xylene or chloroform were mixed by vortexing and incubated at 30 ± 2 °C for 1hr for phase separations. The aqueous phase was taken gently to measure its absorbance at 600 nm and calculated the % of hydrophobicity using the following calculation.

$$= 100 \times (ODinitial - ODfinal)/ODinitial = \Delta OD/ODinitial \times 100 \tag{1}$$

### 2.7. Organic Acid Production in MRS Broth by KCC-37 and KCC-38

Fresh culture was inoculated in MRS broth and incubated at 37 °C for 48 h and cell-free supernatant was separated by centrifugation and used to determine the fermentative organic acids such as lactic acid, acetic acid and succinic acid using the megazyme assay kits (Bray Co. Leinster Wicklow, Ireland).

### 2.8. Inoculums Preparation and Low Moisture IRG Silages Production

The KCC-37 and KCC-38 were cultured in MRS broth (CONDA, Madrid, Spain) for 24 h. Pellets were collected by centrifugation at 4000× *g* for 30 min at 4 °C. Pellets were washed with PBS (GIBCO) thrice and diluted in the same buffer. The LAB colonies were enumerated by Quantom Tx Microbial cell counter with Quantom total cell staining kit (Logos biosystem, Suwon, Gyeonggi-do, South Korea) and used for silage production [2].

Italian ryegrass powder was obtained from the National Institute of Animal Science, Seonghwan-eup, Cheonan, South Korea and prepared samples with low moisture content (35–40%) using sterile distilled water under sterilized condition [2]. A hundred gram of sample was taken in polythene bag size 28 × 36 cm (Aostar Co., Ltd., Seoul, Korea) and categorized into five groups, each consist of three replicates. Group I is the control without inoculums; in group II samples were treated with KCC-37($10^5$), in group III samples weretreated with KCC-38 ($10^5$), in group IV samples were treated with co-culture of KCC-37 and KCC-38, Group V samples treated with standard *L. plantarum* at density of $1 \times 10^5$ (KACC-91067). The air was evacuated from the bags using a vacuum sealer (Food saver V48802, MK Corporation, Seoul, Korea). All bags were prepared and stored for 30 days in the laboratory condition (30 ± 2 °C). After fermentation, the bags were opened for further analysis.

#### 2.8.1. Fermentative Profile

Water extracts of silage samples were prepared by weighing 20 g of silage in 80 mL deionized water and kept in shaking incubator at 150 g (Vision Scientific Co Ltd., Daejeon-Si, Korea) for 60 min at

4 °C. Then all samples were centrifuged at 4000× *g* at 4 °C for 20 min. The aqueous extract was divided into two portions: one portion was used to measure the pH using a combination electrode (inolab pH meter, Thomas Scientific, Swedesboro, New Jersey, USA) and the fermentative organic acids such as lactic acid, acetic acid and succinic acid were determined by the megazyme assay kits (Bray Co. Leinster, Wicklow, Ireland).

### 2.8.2. Quantification of Microbial Populations

The second portion of aqueous extracts was used to quantify the LAB, Yeast and fungi population. The bacteria colonies were enumerated by Quantom Tx Microbial cell counter with Quantom total cell staining kit (Logos biosystem, Gyeonggi-do, Suwon, South Korea). Cultivations of fungi and yeast were performed in potato dextrose agar medium (BD Difco, New Jersey, USA) and petrifilm (3M Microbiology Products, Maplewood, MN, USA), respectively.

### 2.9. Statistical Analysis

Data analysis was carried out using SPSS software (SPSS-16 Inc., Chicago, IL, USA), all numerical data from three independent experiments were analyzed (One Way Anova, Multivariate analysis with duncan). The results were presented as mean ± standard of error of the mean ($p < 0.05$).

## 3. Results

### 3.1. LAB Isolation and Molecular Screening

The LAB strains KCC-37 and KCC-38 were isolated depend upon their distinct morphology from the alfalfa sample. The basic physiochemical test showed that the isolated strains were gram-positive, catalase-negative, non-motile, rod shape (Figure 1A,B) and mesophilic. We then isolated strains were subjected to molecular characterization by 16SrRNA sequencing, antifungal, and probiotic characteristics studies. 16SrRNA sequences revealed that selected strains KCC-37 and KCC-38 belonged to *Lactobacillus plantarum*. The sequences of both isolates were submitted to the NCBI database (Accession number: KP091753 and KP091754).

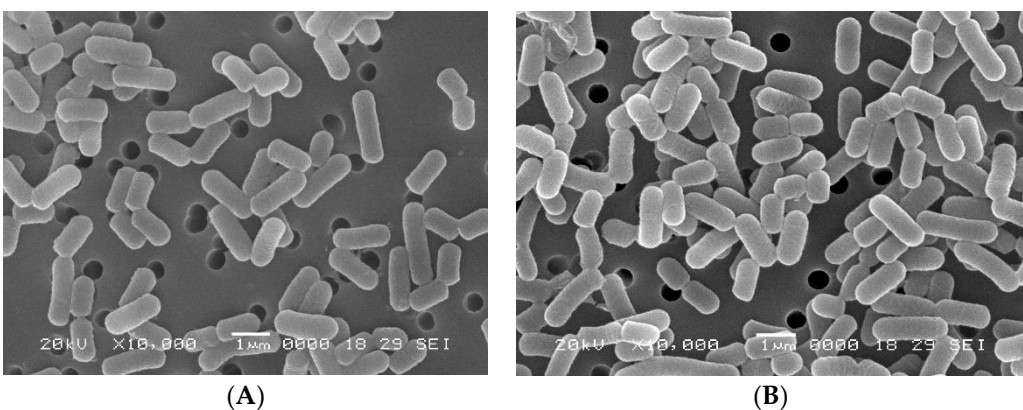

**(A)**   **(B)**

**Figure 1.** Morphology of *L. plantarum* KCC-37 and *L. plantarum* KCC-38. Morphological characteristic were captured by the SEM image system at 10,000 × magnification. (**A**) *L. plantarum* KCC-37; (**B**) *L. plantarum* KCC-38.

### 3.2. Biochemical Characterization of L. plantarum KCC-37 and L. plantarum KCC-38

The *L. plantarum* showed significant ability to ferment the different carbohydrate substrates such as glucose, maltose, Sorbose, Rhamnose, Arabinose and Xylose etc. (Table 1). In addition, both strains have the strong efficiency to produce the various industrially valuable extracellular enzymes, among the productions, alkaline phosphatase, lipase, leucine arylamidase, Naphthol-AS-biphosphohydrolase,

β-galactosidase, α-galactosidase, *N*-acetyl–β glucosaminidase, α-Mannosidase and α-Fucosidase were actively produced by both strains (Table 2).

**Table 1.** KCC-37 and KCC-38 efficiencies on fermentation of different carbohydrates.

| S.No | Name of the Carbohydrates | KCC-37 | KCC-38 |
|:---:|:---:|:---:|:---:|
| 1 | Glycerol | - | - |
| 2 | Erythritol | + | + |
| 3 | D-Arabinose | + | + |
| 4 | L-Arabinose | + | + |
| 5 | D-Ribose | + | + |
| 6 | D-Xylose | + | + |
| 7 | L-Xylose | + | + |
| 8 | D-Adonitol | - | - |
| 9 | Methyl-βD-Xylopyranoside | - | - |
| 10 | D-Galactose | + | - |
| 11 | D-Glucose | + | + |
| 12 | D-Fructose | - | + |
| 13 | D-Mannose | + | - |
| 14 | L-Sorbose | + | + |
| 15 | L-Rhamnose | + | + |
| 16 | Dulcitol | + | + |
| 17 | Inositol | + | + |
| 18 | D-Mannitol | + | + |
| 19 | D-Sorbitol | + | + |
| 20 | Methyl-αD-Mannopyranoside | + | + |
| 21 | Methyl-αD-Glucopyranoside | + | + |
| 22 | N-Acetylglucosamine | - | - |
| 23 | Amygdalin | + | + |
| 24 | Arbutin | - | - |
| 25 | Esculinferric citrate | + | + |
| 26 | Salicin | - | + |
| 27 | D-Cellobiose | + | + |
| 28 | D-Maltose | + | - |
| 29 | D-Lactose | + | + |
| 30 | D-Melibiose | - | + |
| 31 | D-Saccharose | + | - |
| 32 | D-Trehalose | + | + |
| 33 | Inulin | - | + |
| 34 | D-Melezitose | + | + |
| 35 | D-Raffinose | + | + |
| 36 | Amidon | - | - |
| 37 | Glycogen | - | + |
| 38 | Xylitol | + | + |
| 39 | Gentiobiose | + | + |
| 40 | D-Turanose | + | + |
| 41 | D-Lyxose | + | + |
| 42 | D-Tagatose | + | + |
| 43 | D-Fucose | + | - |
| 44 | L-Fucose | + | - |
| 45 | D-Arabitol | + | - |
| 46 | L-Arabitol | - | - |
| 47 | Potassium gluconate | + | + |
| 48 | Potassium2-keto gluconate | + | - |
| 49 | potassium 5-keto gluconate | - | + |

+ Positive fermentation, - Negative fermentation.

**Table 2.** Production of extracellular enzymes by *L. plantarum* KCC-37 and *L. plantarum* KCC-38.

| S.No | Extracellular Enzymes | KCC-37 | KCC-38 |
|------|----------------------|--------|--------|
| 1 | Alkaline phosphatase | +++ | +++ |
| 2 | Esterase ($C_4$) | ++ | ++ |
| 3 | Esterase lipase ($C_8$) | + | + |
| 4 | Lipase ($C_{14}$) | +++ | +++ |
| 5 | Leucine arylamidase | +++ | +++ |
| 6 | Valine arylamidase | + | + |
| 7 | Cystine arylamidase | +++ | +++ |
| 8 | Trypsin | + | + |
| 9 | α-Chymotrypsin | ++ | ++ |
| 10 | Acid phosphatase | ++ | ++ |
| 11 | Naphthol-AS-biphosphohydrolase | +++ | +++ |
| 12 | α-Galactosidase | +++ | ++ |
| 13 | β-Galactosidase | +++ | +++ |
| 14 | β-Glucuronidase | ++ | ++ |
| 15 | α-Glucosidase | ++ | ++ |
| 16 | β-Glucosidase | ++ | ++ |
| 17 | N-Acetyl-β-glucosaminidase | +++ | +++ |
| 18 | α-Mannosidase | +++ | +++ |
| 19 | α-Fucosidase | +++ | +++ |

+; Weak production: ++; moderate production: +++; strong production.

## 3.3. Antibiotic Sensitivity Assay

The antibiotic sensitivity patterns of KCC-37 and KCC-38 were tested with commercially available antibiotics. The results were categorized into resistance (R), and susceptibility(S). The KCC-37 and KCC-38 exhibited high sensitivity patterns against tested antibiotics such as chloramphenicol, nitrofurantoin, streptomycin, ampicillin, cefoxitin, dicloxacillin, cefuroxime and showed resistant property against, gentamicin, tetracycline, Amikacin antibiotics (Table 3).

**Table 3.** Antibiotic sensitivity patterns for *L. plantarum* KCC-37 and *L. plantarum* KCC-38 against commercial antibiotics.

| S. No | Name of Antibiotics | µg/disc | KCC-37 | KCC-38 |
|-------|---------------------|---------|--------|--------|
| 1 | Chloramphenicol (C) | 50 | S | S |
| 2 | Kanamycin (K) | 30 | S | R |
| 3 | Nitrofurantoin (NIT) | 50 | S | S |
| 4 | Tetracycline (TE) | 100 | R | R |
| 5 | Streptomycin (S) | 25 | S | S |
| 6 | Sulphafurazole (SF) | 300 | S | S |
| 7 | Colistin methane sulphonate | 100 | S | S |
| 8 | Dicloxacillin (D/C) | 1 | S | S |
| 9 | Ampicillin (AMP) | 10 | S | S |
| 10 | Amikacin (AK) | 30 | R | R |
| 11 | Gentamicin (GEN) | 10 | S | S |
| 12 | Cefoxitin (CX) | 30 | S | S |
| 13 | Cefalexin (CN) | 30 | S | S |
| 14 | Cefuroxime (CXM) | 30 | S | S |
| 15 | Co-Trimoxazole (COT) | 25 | S | S |

R—Resistant against tested antibiotics; S (>10 mm)—Sensitivity against tested antibiotics.

## 3.4. Production of Organic Acids in MRS Broth

KCC-37 and KCC-38 produced a significant amount of fermentative organic acids, including lactic acid, acetic acid, and succinic acid. However, significant differences existed in the production of lactic

acid concentration among the investigated strains. KCC-38 produced a higher amount of lactic acid than the KCC-37 while other acid production was similar in both strains (Table 4).

**Table 4.** Fermentative acids production by *L. plantarum* KCC-37 and *L. plantarum* KCC-38 in MRS broth.

| Organic Acids (mg/mL) | KCC-37 | KCC-38 |
| --- | --- | --- |
| Lactic acid | 56.52 ± 1.65 [b] | 72.32 ± 3.52 [a] |
| Acetic acid | 17.14 ± 3.25 [c] | 17.32 ± 1.02 [c] |
| Succinic acid | 2.523 ± 0.325 [d] | 2.56 ± 0.152 [d] |

Data represent mean ± STD of three replicates. Different letter a, b, c, and d within a column indicates significances differences between KCC-37 and KCC-38 ($p < 0.05$).

### 3.5. Antifungal Activity of KCC-37 and KCC-38

We then determined the antifungal activity of KCC-37 and KCC-38 by pour plate method. Result exhibited that KCC-37 and KCC-38 strongly inhibit the growth of fungi around KCC-37 and KCC-38 colonies in experimental plates. Both bacteria showed maximum inhibitory zones against *A. clavatus* and *F. oxysporum*. The data suggested that all tested fungal growth was inhibited significantly by KCC-37 and KCC-38 inoculated plates (Table 5).

**Table 5.** Antifungal activities of *L. plantarum* KCC-37 and *L. plantarum* KCC-38 against different fungi, screened by pour plate method.

| Tested Fungi | KCC-37 | KCC-38 |
| --- | --- | --- |
| *S. brevicaulis* | 15.46 ± 1.51 [c] | 14.92 ± 1.36 [c] |
| *A. clavatus* | 36.21 ± 0.58 [a] | 30.37 ± 1.07 [a] |
| *P. chrysogenum* | 24.67 ± 1.35 [b] | 21.51 ± 0.54 [b] |
| *A. flavus* | 24.94 ± 1.48 [b] | 22.47 ± 0.66 [b] |
| *F. oxysporum* | 35.03 ± 0.33 [a] | 30.72 ± 1.28 [a] |

Data represent mean ± STD of three replicates. Different letter a, b, and c between columns indicates significances differences between fungi ($p < 0.05$). Zone formation around LAB as mentioned as millimeter (mm).

Fungal biomass inhibitory property of cell-free metabolites (CFM) produced by KCC-37 and KCC-38 at different temperatures and pH against *S. brevicaulis*, *A. clavatus*, *P. chrysogenum*, *A. flavus*, and *F. oxysporum* was determined. Cell-free metabolites of KCC-37 and KCC-38 significantly reduced biomass of fungi in PD broth supplemented with 10% CFM. Cell-free supernatant produced at 35 °C exhibited strong antifungal activity against all tested fungi than the metabolites produced at 25 and 45 °C (Figure 2).

Also, we analyzed fungal biomass inhibition study for CFM produced at different pH environments. Results indicated that CFM produced at pH6.5 and pH7.5 showed strong biomass reduction against all tested fungi than the other pH. This result suggested that KCC-37 and KCC-38 might be produced higher amounts of antifungal metabolites at optimum temperature 35 °C and optimum pH ranges between 6.5 and 7.5 (Figure 3).

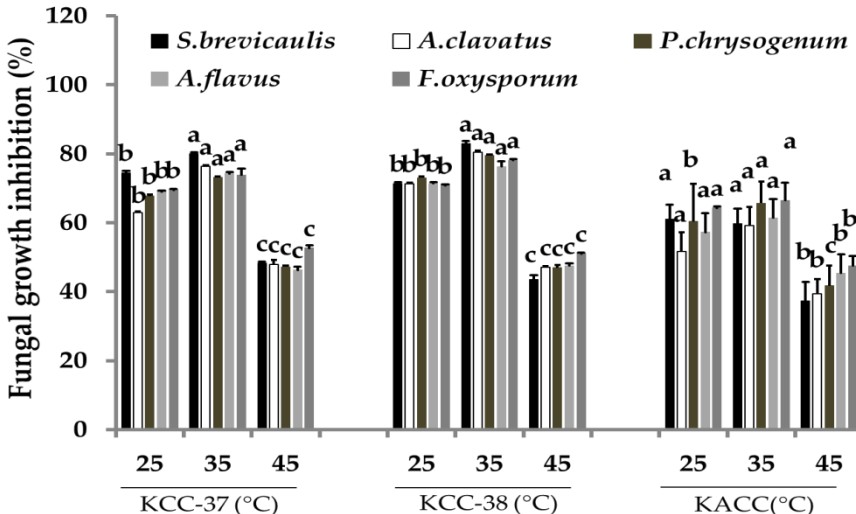

**Figure 2.** Effect of *L. plantarum* KCC-37 and *L. plantarum* KCC-38 CFM produced at different temperatures (°C) on fungal growths (% of fungi growth inhibition, compared to control). Data represent mean ± STD of three replicates. Different letter a, b, and c within a column indicates significances differences between cell-free metabolites (CFM) produced at different temperatures ($p < 0.05$).

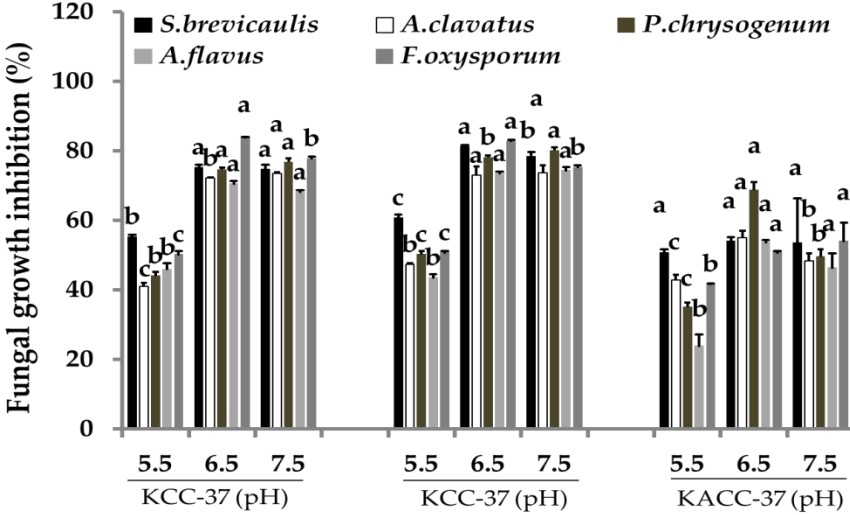

**Figure 3.** Effect of *L. plantarum* KCC-37 and *L. plantarum* KCC-38 CFM produced at different pH on fungal growths (% of fungi growth inhibition, compared to control.) Data represent mean ± STD of three replicates. Different letters a, b, and c within a column indicates significances differences between CFM produced at different pH ($p < 0.05$).

### 3.6. Survival Ability of KCC-37 and KCC-38 in Acidic and Bile Salt Environment

The survival tolerance of the isolated bacterial strains KCC-37 and KCC-38 in different pH environments was analyzed. The increase in acidic environment showed a reduction in colony growth. However, all strains were managed to show better results than the standard *L. plantarum* strain at the highly acidic environment. This result added an advantage to the acid tolerance potential of KCC-37 and KCC-38. Both strains exhibited better resistance at low pH (2.5) conditions (Figure 4A). This was subjected to know the durability in stomach conditions at 37 °C. The tolerance against highly bile salts concentration is considered as the desirable criteria for the potential probiotic strain. In this experiment, the isolated *L. plantarum* KCC-37 and KCC-38 showed significant resistance property against 0.3% oxgall and 0.5% sodium deoxycholate bile slats environment (Figure 4B).

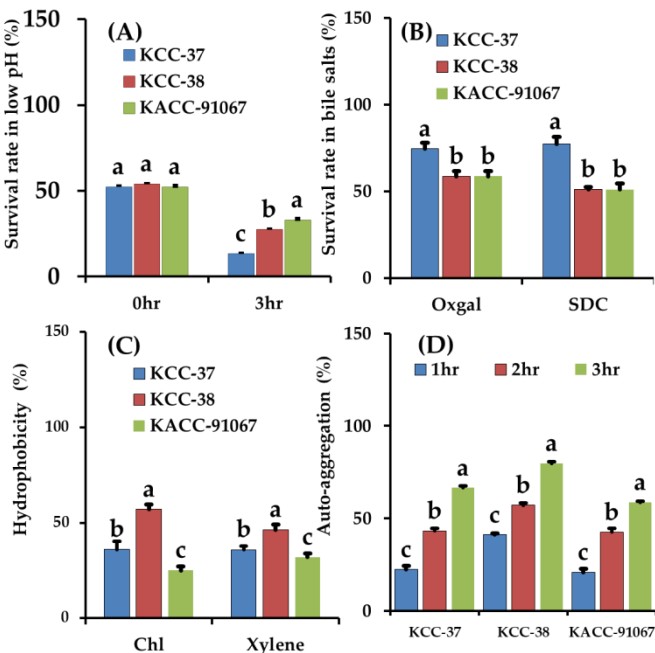

**Figure 4.** Probiotic features of *L. plantarum* KCC-37 and *L. plantarum* KCC-38- In-vitro study. (**A**) The survival rate of *L. plantarum* KCC-37 and *L. plantarum* KCC-38 in acidic conditions. (**B**) The tolerant capability of KCC-37 and KCC-38 in bile salts supplemented MRS broth; (**C**) Hydrophobicity property in hydrocarbons chloroform (Chl) and xylene; (**D**) auto-aggregation features of KCC-37 and KCC-38. The results are expressed as the mean ± STD of three replicates. Different letters a, b, and c within a column indicates significances differences at $p < 0.05$ level.

### 3.7. Auto-Aggregation and Hydrophobicity of KCC-37 and KCC-38

Hydrophobicity of KCC-37 and KCC-38 were determined using hydrocarbons such as chloroform and xylene (Figure 4C). Both strains exhibited a higher percentage of adhesion in the chloroform than the xylene ($p < 0.05$). *L. plantarum* KCC-37 and KCC-38 aggregates significantly on the bottom of tubes in a time- dependent manner (Figure 4D). The turbidity of bacterial suspension was reduced when the incubation periods were extended ($p < 0.05$).

### 3.8. Impact of KCC-37 and KCC-38 on Low Moisture Fermentation of IRG

Finally, we investigated KCC-37 and KCC-38 efficiency on the low moisture silage fermentation. Non inoculated control silage exhibited higher pH level, lower organic acid profiles and less population of LAB in control silages which indicate poor fermentation capability of naturally found LAB in the samples (Figure 5A–C). KCC-37 and KCC-38 inoculated silages exhibited higher amounts of lactic acid with significant amounts of acetic acid and succinic acid. Higher LAB populations were noted in inoculated silage. Fungi and yeast were not detected in both control and inoculated silages. Our inoculums potentially enhanced acidification of silages than the control. Interestingly, co-treatment of KCC-37 and KCC-38 rapidly increased the acidification of silages than the single culture treatments. It also enhanced the production of the lactic acid content of silages ($p < 0.05$).

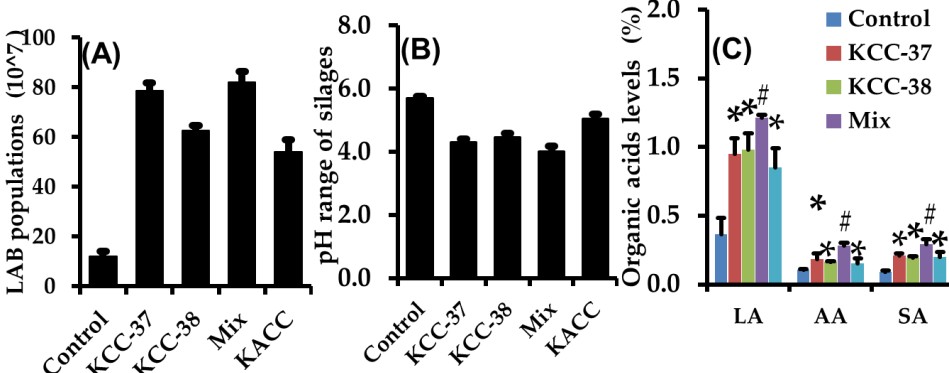

**Figure 5.** Microbial populations, acidification of silages and Organic acids profiles in fermented low moisture IRG silage. (**A**) Microbial population in *L. plantarum* KCC-37, and *L. plantarum* KCC-38 inoculated and non-inoculated low moisture silage; (**B**) pH of the low moisture silages (**C**) organic acids including lactic acid (LA), acetic acid (AA), succinic acid (SA) in the experimental silages. Data represent the mean ± STD (*n* = 3), * Significance between treated and untreated group at *p* < 0.05 level. # Significance between the mixed culture treated, single culture treated and control groups at *p* < 0.05 level.

## 4. Discussion

Beyond technological development, demand is currently raising for the identification of novel lactic acid bacteria (LAB) as potent probiotic candidates [22,23]. In the present study, LAB was isolated from fermented alfalfa samples, identified by biochemical and molecular methods and characterized in-vitro for recognized probiotic properties including acidity, bile salt tolerance and antimicrobial and anti-oxidant activities are key features to consider bacteria as potent probiotics. In the present study gram-positive, catalase-negative, non-sporulated and rod-shaped bacterial strains were isolated and characterized. The primary result confirmed that these bacteria corresponded to the lactobacteriaceae family. These bacteria were able to ferment the different types of carbohydrates. The carbohydrates fermentation ability of these strains varied among the isolated strains. In addition, isolates secreted different type's extracellular enzymes which are closely associated with proteolytic, glycolytic and lipolytic functions. These enzymes might be useful in nutrient digestibility, which provide additional beneficial characteristic to a probiotic. The extracellular enzyme productions have been varied among the LAB [24]. It might be a reason behind the development of species-specific probiotics. Sensitivity to antimicrobial substances is much important criteria for the selection of probiotic bacteria [25]. All isolates were significantly susceptible to antibiotics such as Chloramphenicol, Nitrofurantoin, Streptomycin, and Dicloxacillin etc. The antibiotic susceptible/ resistance is variable with different LAB, some of the LAB had resistance to ampicillin, vancomycin, chloramphenicol, streptomycin, neomycin, nalidixic acid, gentamycin, kanamycin and novobiocin [26]. In the current study, the isolates exhibited resistance against few antibiotics such as Amikacin Tetracycline and Kanamycin. However, KCC-37 and KCC-38 exhibit high sensitivity against most of the commonly available antibiotics.

The secretion of gastric acid in the stomach constitutes a primary defense mechanism. All ingested microorganisms must be overcome these situations, including probiotics [27]. The LAB isolates from alfalfa were screened for their ability to survive in the gastrointestinal tract (GIT) environments. The isolated LAB significantly survived in acidic nature of the stomach and duodenal conditions. Isolated strains showed comparatively lesser survival rates than the standard strain. For the bile salt test, both isolated strains have survived in the bile salts environment of small intestinal conditions. We noted that the KCC-37 survived a higher percentage in bile salt as compared to KCC-38 and standard strain. However, both isolated strains have the ability to cross and survive in GIT environments; it is an essential criterion for the selection of good probiotic bacteria. Cell surface properties by hydrophobicity and autoaggregation are specific parameters for probiotic bacterial adhesion to endothelial cells in the

intestine and can able to function in the gut environment [28]. In the current study, both *L. plantarum* showed good hydrophobicity property towards hydrocarbons chloroform and xylene and all strains exhibited significant autoaggregation features. However, significant differences existed among the investigated strains, which may be attributed to differences in hydrophobic and hydrophilic extensions in the cell wall of bacteria. *L. plantarum* KCC-38 possessed a comparatively higher percentage of hydrophobicity and aggregations properties compared to other strains KCC-37 and KACC-91067.

We then analyzed the antifungal activity of selected Lactic acid bacteria (LAB). The numbers of LAB from various origins show strong antifungal activity and bio-preservation potential [29,30]. Especially *Lactobacillus* sp. is the most prevalent bacterial isolates associated with antifungal activity. Among the species, *L. palntarum* is considered as the most well-known LAB species for its antifungal activity [20,31]. *L. plantarum* exhibited antifungal activity against different fungal species including *Aspergillus* sp., *pencillum* sp. and other fungal spices [32,33]. Current study, antifungal activity of *L. palntarum* KCC-37 and KCC-38 studied against *S. brevicaulis A. clavatus*, *P. chrysogenum*, *A. flavus*, and *F. oxysporum*. Cell-free metabolites produced at different temperatures and pH exhibits potent antifungal activity against all tested fungi. This antifungal activity varied with CFM produced by KCC-37 and KCC-38 at different conditions. In general, optimum temperature and pH influence the production of antimicrobial metabolites. In our experiment, CFM of both strains showed strong antifungal activity while KCC-37 and KCC-38 grown in temperature at 35 °C and pH at 6.5. LAB inhibits the growth of fungi through the production of organic acids. Among the organic acid produced by LAB, the lactic acid is considered as major metabolites, usually produced higher amounts compared to other organic acids. LAB produced a significant amount of acetic acid and propinoic acid that inhibits the growth of fungi [34]. In the current study we noted, KCC-37 and KCC-38 produced a higher amount of lactic acid with a significant amount of acetic and succinic acid in MRS broth. It might be one of the reasons behind the antifungal activity of KCC-37 and KCC-38 against all tested fungi [20,35].

*L. plantarum* KCC-37 and KCC-38 showed effective inhibition against different fungi, but significant differences were noted in inhibitory effects of CFM produced at different temperatures and pH environments. We then investigated the role of *L. plantarum* KCC-37 and KCC-38 in the fermentation of Italian ryegrass at low moisture conditions. An addition of LAB as inoculants at ensiling is intended to ensure rapid and vigorous fermentation that results in rapid acidification and higher amounts lactic acid production at the early stage of ensiling and inhibiting the growth of undesirable bacteria [2,36,37]. In general epiphytic LAB converts the water-soluble carbohydrates into organic acids in the samples during ensilation, which reduced pH of the silages and also preserved for a long time. LAB population in forage samples are very less, it's not sufficient to initiate the fermentation rapidly and also unable to produce a higher concentration of lactic acid during fermentation, this condition could favor the growth of clostridia and causes deterioration of silage quality [38]. Many reports claimed that an addition of *L. plantarum*, *L. pentosus*, and *Pediococcus pentosaceus* effectively ferment the different types of grass and legume plants and preserve its life for a long time [39–41]. Therefore, the addition of LAB is essential criterion for enhancing the fermentation process and rapidly decreased the pH of the silage by producing a higher amount of lactic acid with a lesser extent to acetic acid. Due to the production of these acids, the pH of the ensiled silages decreases and spoilage microorganisms are inhibited. LAB rapidly prevents deterioration of silage quality for a long time [42,43]. Current study, we used *L. plantarum* KCC-37 and KCC-38 as inoculants for low moisture silage production. Both strains rapidly acidified silages than the non-inoculated silages. It is due to higher numbers of LAB noted in inoculated silages than the control. It confirmed that KCC-37 and KCC-38 play a major role in the reduction of pH of the silages by producing high level of lactic acid with marginal amounts of other organic acids whereas in control silages exhibited low numbers of LAB with less concentration of lactic acid, therefore control silages showed higher in pH than the inoculated silages. KCC-37 and KCC-38 combined treatment increased all organic acid production and lowered pH of the silages than the single culture treatments, which indicates synergistically work together and produced a higher amount of organic acids. This data suggested that the *L.plantarum* KCC-37 and KCC-38 efficiently

increased fermentation of Italian ryegrass silage at low moisture conditions. Also, improved quality by increasing lactic acid contents in samples, compared to non-inoculated silage.

## 5. Conclusions

*L. plantarum* KCC-37 and KCC-38, isolated from alfalfa exhibited effective antifungal activity against tested fungi by producing various organic acids. Also, KCC-37 and KCC-38 were survived in the harsh conditions of the gastrointestinal tract including low pH and bile salts environment. These strains showed strong hydrophobicity and aggregations properties that effectively prevent pathogen adhesion to intestinal cells. KCC-37 and KCC-38 enhanced acidification via producing high lactic acid content with a lesser extent to acetic acid and succinic acid in low moisture Italian ryegrass (IRG) at the ensiled condition. These acids decrease the pH of ensiled silage and spoilage microbial growth was inhibited. Mixed strains were more effectively involved in silage fermentation than the single strain inoculated silage. It confirmed these strains synergistically work together and improved low moisture silage quality. Overall suggest that the *L. plantarum-* KCC-37 and *L. plantarum*–KCC-38 could serve as the potent fungicidal, probiotic and good preservatives for improving silage quality that could be considered as most effective antifungal strains with potent silage inoculums for animal feed development

**Author Contributions:** Conceptualization, I.S. and K.C.C.; Data curation, K.M. and S.S.; Formal analysis, D.K. and K.D.L.; Funding acquisition, K.C.C.; Investigation, K.M., I.S., S.S. and P.K.; Project administration, K.C.C.; Writing—original draft preparation I.S.; Writing—review & editing, D.K., P.K. and K.D.L. All authors have read and agreed to the published version of the manuscript.

**Funding:** This work was performed with the support of the Cooperative Research Program for Agriculture Science and Technology Development (Project Title: Development of Quality Improvement and Standardization Technique for Low Moisture Storage Forage, Project No PJ01091601), Rural Development Administration, Republic of Korea. This study was supported by a Postdoctoral Fellowship Program of the National Institute of Animal Science, Rural Development Administration and Republic of Korea.

**Acknowledgments:** We thank National Institute of Animal Science, Seonghwan, Cheonan, Korea, for their Laboratory and Technical supports.

**Conflicts of Interest:** All the authors claim that there are no conflicts of interest in this study.

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
