# Peer review of "Probiotic Characteristics and Antifungal Activity of Lactobacillus plantarum and Its Impact on Fermentation of Italian Ryegrass at Low Moisture"

_applsci, doi:10.3390/app10010417_

Round 1
Reviewer 1 Report
The communication has been explored the antifungal activity of Lactobacillus Plantarum and its impact on fermentation of Italian Ryegrass at low moisture. The scientific quality, relevance and presentation of this manuscript is acceptable and can be improved further. Some specific comments for improvement are as follow:
The abstract is too general and should be more specific to the research findings and data. Results section, should add the scanning electron micrograph of Lactobacillus plantarum KCC-37 and 360 KCC-38. What are the antifungal activity of KCC-37 and 360 KCC-38, comparing with others strains? Besides the fermentative organic acids. Are there any other bioactive compounds show antifungal activity? Authors should use statistics to analyze data. (Table 5, 6, and 7) English should be improved, especially in the results section. Authors should double check the format in the manuscript, such as hours or hrs, 4000g or 4000 g, 20min or 20 min, 20 ml or 20 mL…etc. There are other typographical and grammatical errors not specified here. What is the optimization conditions for Lactobacillus plantarum fermentation Italian ryegrass in low moisture?
Author Response
Thank you very much for the reviewer for given valuable comments and suggestions about our manuscript. A lot of comments have given by reviewers about our research paper would improve a quality manuscript. We carefully considered reviewers comments that coloured the revisions made in the manuscript. Language of the manuscript has been checked thoroughly by the language expert and Grammarly software. The paper has been modified accordingly with relevant changes appended separately. We do hope that the revised manuscript is now suitable for publication in Applied Sciences. We look forward to your kind consideration
Responses to reviewer comments
(Comments)The abstract is too general and should be more specific to the research findings and data. Results section, should add the scanning electron micrograph of Lactobacillus plantarum KCC-37 and 360 KCC-38. What are the antifungal activity of KCC-37 and 360 KCC-38, comparing with other strains? Besides the fermentative organic acids. Are there any other bioactive compounds show antifungal activity? Authors should use statistics to analyze data. (Table 5, 6, and 7) English should be improved, especially in the results section. Authors should double-check the format in the manuscript, such as hours or hrs, 4000g or 4000 g, 20min or 20 min, 20 ml or 20 mL…etc. There are other typographical and grammatical errors not specified here. What are the optimization conditions for Lactobacillus plantarum fermentation Italian ryegrass in low moisture?
(Responses) Thank you for your valuable comments and suggestions. We have modified the whole abstract in a concise manner and included key research finding data. We have included the new images of Lactobacillus plantarum taken by Scanning Electron Microscopy in Fig.1
Initially, we screened antifungal activity of different strains isolated from alfalfa by pour plate method. This revealed that two strains showed more potent anti-fungal activities against tested fungi than other strains.
We then analyzed the antifungal efficiency of their cell-free supernatant by fungal biomass study. In this experiment, we used standards L.plantarum obtained from Korean Agricultural Culture Collection (KACC) for comparative study.
Thank you for valuable comments; Lactic acid bacteria (LAB) produced different types of antimicrobial agents which include organic acids, short-chain fatty acids, peptides and others. More importantly, it produces higher amounts of lactic acid than the others. Also, production of lactic acid with marginal amounts of acetic acid and succinic acid in silages during ensiling is essential for making good quality silages.it is the reason for limited quantification of other metabolites. These acids can act as good antimicrobial agents; therefore, we interpreted these acids. In future, we can try to analyses other antimicrobial agents as per reviewer suggestion.
We have included new statistical analysis significant for experimental data of Table, 5, 6 and 7.
Thank you lots; as per suggestion we have modified the units in the whole manuscript according to author guidelines
Sorry for these careless mistakes: All grammar and spell mistakes were carefully checked the whole manuscript and made the red colour
Air was evacuated from experimental silage of control and LAB inoculated bags using a vacuum sealer (Food saver V48802, MK Corporation, Seoul Korea). We used 24hrs LAB as inoculants for silage production. All bags were prepared and stored for 30days in the laboratory condition at 30±2C in anaerobic condition. After 30days, the bags were opened for further analysis.
Reviewer 2 Report
In general, a valuable and well written microbiological and chemical work. Poorly written (typos, bad tables and figures). After a thorough rewording of this text and referring to my prophets - it is suitable for publication.
Abstract:
- All abbreviations must be explained in the text at least once.
Page 1. Line 23. (“Medicago sativa” L.) - italization
Page 1. Line 27: „16srRNAsequence” -- > 16S rDNA sequence
Page 1. Line 27: “Lactobacillus Plantarum” -- > Please write the species name in lowercase (change this throughout the manuscriptt).
Page 1. Line 31-33. Please describe if lactic acid also correlated with lowering the pH. This is the most important acid.
Introduction
Page 2. Line 55-82 - Only Clostridium was mentioned, and where about the fungi? Since undesirable fungi are also analyzed, you should mentioned this in the introduction.
Suggestion for pargraph: “The most common fungi that cause silage contamination and produce mycotoxins belong to the species Fusarium, Penicilium and Aspergillus. Undesirable fungi can come from bioaerosol or exist as plant epiphytes or endophytes.” (Alonso et al. 2013, Przemieniecki et al. 2019, FijaÅ‚kowska et al. 2019)
Alonso VA, Pereyra, CM, Keller LA, Dalcero AM, Rosa CA, Chiacchiera SM and Cavaglieri LR. Fungi and mycotoxins in silage: an overview. J Appl Microbiol 115:637–643, 2013.
Przemieniecki, S.W.; Damszel, M.; Kurowski, T.P.; Mastalerz, J.; Kotlarz, K. Identification, ecological evaluation and phylogenetic analysis of non-symbiotic endophytic fungi colonizing timothy grass and perennial ryegrass grown in adjacent plots. Grass Forage Sci. 74(1):42-52. 2019
Fijałkowska, M., Przemieniecki, S.W., Purwin, C., Lipiński, K., Kurowski, T.P. and Karwowska, A. (2019), The effect of an additive containing three Lactobacillus species on the fermentation pattern and microbiological status of silage. J Sci Food Agric. doi:10.1002/jsfa.10126
Page 2. Line 77. Please added hypothesis.
Materials and methods
- Some methods are not sufficiently described. Please describe the protocols so that they can be reproduced or provide the citation of the methodology.
- It is good practice to provide the name of the manufacturer of equipment, reagents and media. You didn't do it, or you did it in different way. Please correct it.
- Please read the rules for placing mathematical formulas in a manuscript.
Page 3. Line 86: “1g sample was” – A one gram of sample was…
Page 3. Line 91: What incubation temperature was used? For environmental Lactobacillus spp. it is 30 degrees C, and for symbionts it is 36 degrees C.
Page 3. Line 96-97: Citation needed for primers and PCR:
Lane, D. 16S/23S rRNA sequencing. In Nucleic Acid Techniques in Bacterial Systematics; John Wiley & Sons: New York, NY, USA, 1991; pp. 115–175.
Page 3. Line 105: KACC? -> what this abbreviation means? Korean Agricultural Culture Collection?
Page 3. Line 99: BLAST need citation Altschul et al.1990
Altschul S, Gish W, Miller W, Myers E, Lipman D. Basic local alignment search tool. J Mol Biol 215:403–410. 1990.
Page 5. Line 176-179, and Figure 2. What statistical test was used? Please describe.
Figure 2: Low legibility of figures. Please modify to make them more aesthetic and legible.
The one major mistake in this work was the lack of quantitative determination of butyric acid (which you also mentioned in the introduction as an indicator of quality).
Page 5. Line 181: “Isolation and molecular screening” -- > LAB isolation and molecular screening
Page 5. Line 188: “NCBI database” -- > GenBank database
Results
- Chapter quite poorly written in terms of observed features.
Page 7. Line 223-225: A bit too poorly described. You can indicate the sum of acids tested in this chapter. And indicated strain with better LA production.
Discussion
- A quite wellwritten chapter. However Too much is written about what was done and too little about the effects of what came out and comparisons with other studies. A reference to works with a detailed description of the LAB's operation and other similar works could be useful. Eg.:
Hager Alhaag, Xianjun Yuan, Azizza Mala, Junfeng Bai and Tao Shao, Fermentation Characteristics of Lactobacillus Plantarum and Pediococcus Species Isolated from Sweet Sorghum Silage and Their Application as Silage Inoculants, Applied Sciences, 10.3390/app9061247, 9, 6, (1247), (2019)
Page 11. Line 323-343: No reference to very important features: temperature and pH tolerance.
Page 11. Line 350: 16srRNA -- > 16S rRNA
Page 12. Line 359: penicillium -- > Penicillium
Page 12. Line 369: “significant amount of acetic and succinic acid” - This has to be compared with literature. The concentrations described as satisfactory are described in other papers.
Page 12. Line 372-391 (and sentence: “Therefore, the addition of LAB is essential criteria for 379 enhancing fermentation process and rapidly decreased pH of the silage by producing a higher 380 amount of lactic acid with marginal amounts of other organic acids. LAB rapidly prevents 381 deterioration of silage quality for long time [29,30].”)
- What bacteria? What happens when they are homo and what happens when they are heterofermentative? Also refer to the strains tested.
- You are discussing Clostridium and you should be talking about undesirable fungi. If you write about Clostridium - prove that it absent.
Conclusion – Quite well written.
“probiotic” - This cannot be highlighted in the conclusions because they have not been tested on animals.
Author Response
Very happy and thank the reviewer for giving huge valuable comments and suggestion about the paper. Comments have provided by reviewers are more useful that would improve a quality manuscript. We carefully considered reviewers comments that coloured the revisions made in the manuscript. Language of the manuscript has been checked thoroughly by the language expert and Grammarly software. The paper has been modified accordingly with relevant changes appended separately. We do hope that the revised manuscript is now suitable for publication in Applied Sciences. We look forward to your kind consideration
Responses to reviewer comments
Comments: Abstract:- All abbreviations must be explained in the text at least once.
Page 1. Line 23. (“Medicago sativa” L.) - italization,
Page 1. Line 27: „16srRNAsequence” -- > 16S rDNA sequence
Page 1. Line 27: “Lactobacillus Plantarum” -- > Please write the species name in lowercase (change this throughout the manuscript).
Responses: Thank you for your valuable suggestions; we have carefully checked and modified accordingly in the whole manuscript. All changes have made of red colour.
Comment: Page 1. Line 31-33. Please describe if lactic acid also correlated with lowering the pH. This is the most important acid.
Responses :Yes, we have mentioned essential of lactic acid in abstract as well as in introduction parts.
In an anaerobic condition, lactic acid bacteria (LAB) utilize water-soluble carbohydrates (WSC) and converted into organic acids particularly lactic acid that gradually decreases pH the surrounding environment during the fermentation process, is known as ensiling. Enhancement of lactic acid production is an essential factor for developing high-quality silage and used to preserve it’s for a long time (Line No:45-49)
Comment : Page 2. Line 55-82 - Only Clostridium was mentioned, and where about the fungi? Since undesirable fungi are also analyzed, you should mention this in the introduction.
Responses : We strongly agreed with reviewer comments and included other fungi which involved in spoilage of silage quality as fungal spoilage and mycotoxin contaminations are one of the highest risks of stored feed such as silage. It affects nutrients composition, dry matter, palatability reduction and reduction in consumption of silage. The main fungi isolated from contaminated silages were Aspergillus, Penicillium and Fusarium genera followed by mucor, and others[4-6]. Homofermentative Lactobacilli includes Lactobacilli casei and L. plantarum that directly enhance the silage/haylage quality by preventing undesirable microbial growth via lowering surrounding pH. (Line No:58-63)
Comment: Page 2. Line 77. Please added hypothesis.
Responses : Yes, as per the reviewer suggestion, we have included the hypothesis of this experiment at the end of the introduction section (Line No: 83-90)
Comments : Materials and methods
-Some methods are not sufficiently described. Please describe the protocols so that they can be reproduced or provide the citation of the methodology. It is good practice to provide the name of the manufacturer of equipment, reagents and media. You didn't do it, or you did it do diferently. Please correct it.
Responses: Thank you for your valuable suggestions, we have modified according to reviewer suggestion about chemicals, medium and equipment’s used in this study with company name, model and their locations.
Comment: Please read the rules for placing mathematical formulas in a manuscript.
Responses: Sorry for this inconvenience caused and thank you lot for this suggestion, now we have used Microsoft equation three tools to insert mathematical formulas
Comment: Page 3. Line 86: “1g sample was” – one gram of sample was…
Responses: Yes, we have modified as “A one-gram sample was diluted with 10 ml sterile distilled water, and then made serial dilution”.
Comment: Page 3. Line 91: What incubation temperature was used? For environmental Lactobacillus spp. It is 30 degrees C, and for symbionts, it is 36 degrees C.
Responses: Diluted samples were spread on De Man Rogosa and Sharpe Agar (MRS agar, CONDA, Madrid, Spain). Plates were incubated at 30°C for 72hrs. LAB colonies were picked up and streaked on Bromo cresol purple agar medium (BCP, CONDA, Madrid, Spain) to confirm the LAB identity (Line No: 93-98)
Comments: Page 3. Line 96-97: Citation needed for primers and PCR:
Lane, D. 16S/23S rRNA sequencing. In Nucleic Acid Techniques in Bacterial Systematics; John Wiley & Sons: New York, NY, USA, 1991; pp. 115–175.
Page 3. Line 99: BLAST need citation Altschul et al. .1990
Altschul S, Gish W, Miller W, Myers E, Lipman D. Basic local alignment search tool. J Mol Biol 215:403–410. 1990.
Responses: Yes, we have cited the references according to reviewer suggestion (Reference Nos: 14-16)
Comments: Page 3. Line 105: KACC? -> what this abbreviation means? Korean Agricultural Culture Collection?
Responses: Abbreviation KACC is Korean Agricultural Culture Collection. Now we have included in method section (Line No 116-118)
Comments: Page 5. Line 176-179 and Figure 2. What statistical test was used? Please describe.
Responses: We have included a detail of parameters included during analysis (Data analysis was carried out using SPSS software (SPSS-16 Inc., Chicago, IL, USA), all numerical data from three independent experiments were analyzed (One Way Anova, Multivariate analysis with Duncan). The results were presented as mean ± standard of error of the mean (P < 0.05).
Comments: Figure 2: Low legibility of figures. Please modify to make them more aesthetic and legible.
The one major mistake in this work was the lack of quantitative determination of butyric acid (which you also mentioned in the introduction as an indicator of quality).
Responses: We strongly agreed with the reviewer concept and thanks for it. Butyric acid at a high level is considered as negative indicators, and the lactic acid at a higher level is considered as positive indicators of silage quality. At same quantification butyric acid in samples also important for silage production. we will consider reviewer suggestion in the future experimental analysis because currently, we don’t have the kit for butyric acid quantification
Comments: Page 5. Line 181: “Isolation and molecular screening” -- > LAB isolation and molecular screening Page 5. Line 188: “NCBI database” -- > GenBank database
Responses: We have modified above said mistakes in the manuscript (Line Nos:112 and 207)
Comments: Results
- Chapter wrote in terms of observed features.
Page 7. Line 223-225: A bit too poorly described. You can indicate the sum of acids tested in this chapter. And indicated strain with better LA production.
Responses: Thanks for your comments, we have modified and improved the result section of the manuscript according to reviewer comments.
Comments: Discussion
- A quite well-written chapter. However Too much is written about what was done and too little about the effects of what came out and comparisons with other studies. A reference to works with a detailed description of the LAB's operation and other similar works could be useful. Eg.:
Hager Alhaag, Xianjun Yuan, Azizza Mala, Junfeng Bai and Tao Shao, Fermentation Characteristics of Lactobacillus Plantarum and Pediococcus Species Isolated from Sweet Sorghum Silage and Their Application as Silage Inoculants, Applied Sciences, 10.3390/app9061247, 9, 6, (1247), (2019)
Responses: Thanks for your comments, we have modified and improved the discussion part with new references according to reviewer comments.
Reviewer 3 Report
In the present study authors claim to have studied the probiotic potential and antifungal activity of Lactobacillus plantarum and its impact on fermentation of Italian Ryegrass. It is a very interesting study though many grammatical and spelling errors in it is essential that the manuscript is revised by a native English speaker. If not revised many aspects cannot be explained properly. In addition, some major concerns need to be addressed prior publication.
Major concerns:
Authors claim to have isolated potentially probiotic strains. How does this study verify that these strains can be potentially probiotic? The viability during simulation of GIT does nor provide such a claim. Do authors think that they can claim probiotic character by this study? Please explain and revise accordingly. Why did authors study the production of organic acids in MRS broth? Why not test the production of acids in milk? Line 115: “Conidial fungi” Authors do not explain which fungi they applied in order to test the antifungal activity. In Table 5 a list of the fungi is presented but author do not provide the origin of these microorganisms. Why were they selected? Were they sufficient? Was this based on a previous method?
Minor concerns:
Title: Please be carful with microorganism and revise as “Lactobacillus plantarum”. Check this manner through the whole manuscript.
Line 19: “quality food with safety” Please revise
Lines 20-21: Do you mean in the present study? Please check grammar of the whole manuscript.
Line 21: Replace “occurred” with “occurring”
Line 23: Please add a brief explanation about Alfalfa (Medicago sativa)
Line 23: Revise “and screened their antifungal activity” as “and screened for antifungal activity.”
Abstract section: Authors have named the newly isolated strains as Lactobacillus plantarum KCC-37 and Lactobacillus plantarum KCC-38. Please add the strain when you first mention the isolates.
Lines 25-30: Authors should avoid mentioning the names of the strains and just provide their characteristics. For example, the two newly isolated strains were able to survive simulated GIT conditions, exhibit significant hydrophobicity and aggregation properties with intense antifungal activity.
Lines 120-121: “Conidial spore of each fungus was inoculated” which fungus???
Figure 1: Probiotic potential? Please explain the caption of the Figure in detail. Avoid phrases such as probiotic claim of not approved by the results.
Author Response
Very happy and thank the reviewer for giving huge valuable comments and suggestion about the paper. Comments have provided by reviewers are more useful that would improve a quality manuscript. We carefully considered reviewers comments that coloured the revisions made in the manuscript. Language of the manuscript has been checked thoroughly by the language expert and Grammarly software. The paper has been modified accordingly with relevant changes appended separately. We do hope that the revised manuscript is now suitable for publication in Applied Sciences. We look forward to your kind consideration
Comments: Authors claim to have isolated, potentially probiotic strains. How does this study verify that these strains can be potentially probiotic? The viability during simulation of GIT does not provide such a claim. Do authors think that they can claim probiotic character by this study? Please explain and revise accordingly. Why did authors study the production of organic acids in MRS broth? Why not test the production of acids in milk? Line 115: “Conidial fungi” Authors do not explain which fungi they applied to test the antifungal activity. In Table 5, a list of the fungi is presented, but author does not provide the origin of these microorganisms. Why were they selected? Were they sufficient? Was this based on a previous method?
Responses: Thank you for valuable comments and suggestions, we agreed with reviewer suggestion about probiotic potential clam. We analyzed probiotic characteristics of isolated LAB (low acidic, bile salts, hydrophobicity and aggregations properties). It is the preliminary qualification for considering bacteria as probiotics. In this aspect, we claimed that our LAB could be considered s probiotics. However, we have replaced the words “probiotic potential by probiotic characteristics: according to reviewer suggestion.
We mainly have to use these bacteria for silage or haylage production from grass/ legumes plants. Here lactic acid production is an essential factor for making quality silage. In this aspect, we quantified in organic acids, particularly lactic acid in MRS as well as in fermented silages.
List and order of fungi have given in method section now according to your suggestion (Line No 131-134)
Most of the fungi used in the study are involved in the production of deleterious effects on the silage quality except Scopulariopsis brevicaulis
Comments: Minor concerns:
Title: Please be carful with microorganism and revise as “Lactobacillus plantarum”. Check this manner through the whole manuscript.
Line 19: “quality food with safety” Please revise
Lines 20-21: Do you mean in the present study? Please check grammar of the whole manuscript.
Line 21: Replace “occurred” with “occurring.”
Line 23: Please add a brief explanation about Alfalfa (Medicago sativa)
Line 23: Revise “and screened their antifungal activity” as “and screened for antifungal activity.”
Abstract section: Authors have named the newly isolated strains as Lactobacillus plantarum KCC-37 and Lactobacillus plantarum KCC-38. Please add the strain when you first mention the isolates.
Lines 25-30: Authors should avoid mentioning the names of the strains and just provide their characteristics. For example, the two newly isolated strains were able to survive simulated GIT conditions, exhibit significant hydrophobicity and aggregation properties with intense antifungal activity.
Responses: Thanks for giving a lot of comments and suggestions. We have modified the whole abstract according to other reviewers. In the new abstract we have avoided all those mistakes as you mentioned
Comments: Lines 120-121: “Conidial spore of each fungus was inoculated” which fungus???
Responses: Yes now we have included details of conidial fungi in method section (Line No:131-132)
Comments: Figure 1: Probiotic potential? Please explain the caption of the Figure in detail. Avoid phrases such as the probiotic claim of not approved by the results.
Responses: Yes, we agreed and modified accordingly (Line No: 302-303)
Round 2
Reviewer 3 Report
Authors have extensively revised their manuscript. I can recommend publication after some minor check in spelling and grammar.
Minor points
Keywords: Are organic acids a representative keyword of the present study? Please revise accordingly.
Introduction: 1. Authors need to provide more recent data regarding the use of newly isolated probiotic lactic acid bacteria in the production of cheese products. Preserving the efficacy of probiotic bacteria exhibits paramount challenges that need to be addressed during the development of functional dairy products. The significance of appropriate strategies regarding the improved viability and delivery of probiotic bacteria must be highlighted. Also authors can add some more comments regarding the spontaneous fermentation by lactic acid bacteria.
Why newly isolated "wild" stains can be of major importance? Please add your comments in the introduction section as well as in the discussion section.
Lines 347-348: Please add the appropriate reference.
Fig. 1: Please provide the magnification details in the Figure caption not on the Figure.
Fig. 4: Please add the whole name of the strain in the Figure caption.
Fig. 5: Please provide in detail the specific strain represented in each Figure. Also please revise all Figure as analysis is not good. In Figure 5A please replace colony with "viability".
Author Response
Responses to reviewer comments
Very happy and thank the reviewer for giving huge valuable comments and suggestion about the paper. Comments have provided again by the reviewer are more useful that would improve a quality manuscript. We carefully considered reviewers comments that colored the revisions made in the manuscript. Language of the manuscript has been checked thoroughly by the language expert and Grammarly software. The paper has been modified accordingly with relevant changes appended separately. We do hope that the revised manuscript is now suitable for publication in Applied Sciences. We look forward to your kind consideration.
Comment: Keywords: Are organic acids a representative keyword of the present study? Please revise its accordingly.
Response: Yes, we agreed with reviewer comment and was deleted keyword organic acids from the abstract keywords list
Comment: Authors need to provide more recent data regarding the use of newly isolated probiotic lactic acid bacteria in the production of cheese products. Preserving the efficacy of probiotic bacteria exhibits paramount challenges that need to be addressed during the development of functional dairy products. The significance of appropriate strategies regarding the improved viability and delivery of probiotic bacteria must be highlighted. Also authors can add some more comments regarding the spontaneous fermentation by lactic acid bacteria. Why newly isolated "wild" stains can be of major importance? Please add your comments in the introduction section as well as in the discussion section.
Response: Thank you for your valuable suggestions, we have included some details regarding probiotic bacteria and its importance with recent literature according to reviewer suggestion (Line No, 42-51;75-88; 403-406; 416-419; 428-431)
Comment: Lines 347-348: Please add the appropriate reference.
Response: We have cited a reference as per reviewer suggestion reference No: 26
Comment: Fig. 1: Please provide the magnification details in the Figure caption not on the Figure.
Response: Yes agreed with reviewer comment and included same (Line No 216-217)
Comment: Fig. 4: Please add the whole name of the strain in the Figure caption.
Response: Now we have given strains name in all figure legends
Comment: Fig. 5: Please provide in detail the specific strain represented in each Figure. Also please revise all Figures as analysis is not good. In Figure 5A please replace colony with "viability".
Response: Yes, we agreed and modified accordingly